# DENcode: A model for haplotype-informed transmission probability of dengue virus

Sachith Maduranga[1,2], Braulio Mark Valencia[3], Chathurani Sigera[4], Praveen Weeratunga[4], Deepika Fernando[4], Senaka Rajapakse[4], Andrew R. Lloyd[2], Rowena A. Bull[1,2], Haley Stone[5,6], Chaturaka Rodrigo[1,2]*

1 School of Biomedical Sciences, Faculty of Medicine and Health, University of New South Wales, Sydney, New South Wales, Australia, 2 Kirby Institute, University of New South Wales, Sydney, New South Wales, Australia, 3 International Centre for Future Health Systems, University of New South Wales, Sydney, New South Wales, Australia, 4 Faculty of Medicine, University of Colombo, Colombo, Sri Lanka, 5 Computer Science and Engineering, University of New South Wales, Sydney, New South Wales, Australia, 6 University of Glasgow Centre for Virus Research, University of Glasgow, Glasgow, United Kingdom

* c.rodrigo@unsw.edu.au

## Abstract

Dengue virus transmission networks are often only partially resolved, due to gaps in sampling, unobserved mosquito-mediated transmission, and using methods (phylogenetics) that describe evolutionary relatedness but not explicit, probabilistic transmission links between individual infections. We developed DENcode, a framework to estimate the relative likelihood of vector-mediated transmission between pairs of dengue cases by combining a temperature- and time-modulated epidemiological kernel, which captures the extrinsic incubation period and human infectiousness, with a phylogenetically informed genetic similarity kernel derived from patristic distances between viral haplotypes or consensus sequences. Validation with a real-life dataset of 90 dengue infections sampled from Colombo, Sri Lanka between 2017 – 2020 and sequenced to resolve within-host haplotypes, DENcode estimates were stable across 100 Monte Carlo iterations, yielding narrow credible intervals (median width <0.001) and consistent top-ranked transmission pairs. Sensitivity analyses using ablation experiments showed that removing either the genetic or epidemiological component substantially altered the distribution of linkage probabilities, indicating that both contribute meaningfully to the inferred transmission structure. Serotype-specific transmission networks constructed from pairwise linkage probabilities from DENcode were analysed using degree- and path-based centrality measures at probability thresholds of 0.1 and 0.5, revealing relative importance of cases to disease transmission within the community. Haplotype-derived networks were more informative than consensus-based networks (x 3.6 and x 1.6 times more edges for DENV2 and 3 respectively). DENcode is a robust framework to explore dengue transmission within

**Data availability statement:** The raw dengue virus sequence reads used in this study are available under BioProject accession PRJNA1377261. Hotspot analysis data are accessible through https://doi.org/10.1016/j.virol.2025.110750. Location and time metadata are available at https://doi.org/10.3390/v15071408. The DENCode transmission probability dataset, including all P_ij outputs and ablation results can be found at https://doi.org/10.5281/zenodo.17855697. The full DENCode source code and documentation are openly available at https://github.com/DENVGenomics/DENcode.

**Funding:** This work was supported by the University of Colombo, Sri Lanka (grant no, AP/3/2/ 2017/CG/25 to DF) and the National Health and Medical Research Council, Australia (Investigator grant no. 1173666 to CR). The funders had no role in study design, data collection and analysis, decision to publish, or preparation of the manuscript.

**Competing interests:** The authors have declared that no competing interests exist.

a community that provides an output of network of transmission probabilities informed by pathogen genetic similarity and clinical epidemiological parameters.

## Author summary

Tracing epidemics of dengue in setting where dengue transmission happens continuously poses many challenges especially with limited availability of genomic surveillance. Here we introduce a model that uses genomic data together with time and location data to calculate a probability of two cases of dengue being related to each other. Using data from the Colombo dengue study, from 2017 to 2020, Sri Lanka, we evaluated the model. We used haplotype level sequences that correspond to the viral variation within the human host and consensus level sequences that average the data from a single human host into a single sequence. We constructed transmission probability networks for each dengue serotype and were able to identify patients who played key roles in the corresponding networks. We were able to show that this model is robust and will be a valuable tool in the context of dengue control.

## Introduction

Dengue is a flavivirus infection with an estimated 3.97 billion people at risk across 128 countries [1,2]. There are four antigenically distinct serotypes of the dengue virus (DENV1-DENV4) which have 65%-70% nucleotide sequence homology [3]. Infection from one serotype provides lasting immunity against homotypic (same serotype) re-infection [4], but a second heterotypic infection may trigger a more severe illness episode via antibody dependent enhancement (ADE) due to the non-neutralising but cross-reactive antibodies from the first infection [5]. Being a vector borne infection, transmission of dengue is closely linked to the life cycle of its two vector mosquitoes *Ades aegypti* (primary vector) and *Ades albopictus* [6]. Female *Ades aegypti* is an anthropophilic, endophilic day biter that lays eggs in small collections of still water [7]. The estimated average flight distance of *Aedes aegypti* is limited at 106 m (95% CI: 87.68, 123.69) [8] and hence movement of infected people may be the major determinant for the spread of the infection during outbreaks as confirmed by phylogeographic analyses previously. The mean incubation period within the mosquito (Extrinsic Incubation period – EIP) is dependent on the ambient temperature and can range from 5-33 days at 25°C and 2–15 days at 30°C [9].

Most dengue infections are subclinical, and majority of symptomatic cases present as a self-limiting flu-like febrile illness (Dengue Fever). A minority of symptomatic cases (<5%) [10] progress to Severe Dengue defined as severe plasma leakage and/or severe bleeding and/or severe organ impairment [11]. Once the virus enters a human host it takes 5.9 days on average for symptoms to appear (intrinsic incubation period: IIP, 95% CI: 3–10 days). The human host is infective to mosquitoes from

about 2 days before and up to 6 days after the onset of symptoms [12]. Within the human host, the virus replicates and due to the error prone nature of its RNA dependent RNA polymerase, many haplotypes (within-host variants) are generated. These haplotypes are selected by the host immunity based on their fitness [13], and undergoes a diversity bottleneck event (founder effect) when establishing the infection in a new human host. Similar haplotype diversity bottlenecks may occur in the mosquito vector as well [14].

In endemic settings it is often difficult to identify related infections specially during outbreaks where patient numbers increase rapidly within a short time span as a few weeks [15]. Geographical proximity alone is an inadequate measure for this as infected humans move far from the point of infection for work, study and leisure using modern means of transportation during the intrinsic incubation period, sometimes introducing imported dengue infections across countries. Thus, it is important to develop a model that can identify the likelihood of two patients being related to a same infection cluster. Because dengue is an acute infection, the virus has limited opportunity to accumulate mutations within a single human host compared with chronic infections such as HIV or HCV, although low-frequency within-host variants can still arise during the course of infection. Therefore, consensus level sequences of the same serotype (outbreaks are typically dominated by one serotype) and from the same outbreak often does not offer much genetic diversity even when the full-length genome is sequenced. This limitation is potentially avoided if haplotype (within-host variants) level sequences are also considered but this option is less explored due to technical and computational challenges in resolving haplotype sequences. As a result, the true connectivity of dengue infection networks remains poorly resolved, constraining the ability to target interventions and evaluate control policies. Here, we introduce DENcode, a probabilistic framework developed to improve the identification of dengue transmission links in settings where genetic diversity is low and epidemiological signals alone are insufficient. DENcode integrates multiple streams of routinely collected data to provide an estimate of the probability that two infections are connected within the same transmission chain, addressing a long-standing gap in fine-scale dengue transmission inference.

## Overview

DENcode is a probabilistic framework for estimating transmission linkage risk between pairs of dengue patients. It integrates multiple sources of information, including clinical metadata, epidemiological data comprising of dates and locations, climate data, vector specific parameters, and viral genetic similarity (at haplotype or consensus level), into a unified pipeline (Figs 1 and S1).

The model computes the relative likelihood of vector-mediated transmission between a pair of dengue patients by combining two components: a temperature- and time-modulated epidemiological kernel that captures extrinsic incubation period and human infectiousness, and a phylogenetically informed genetic similarity score derived from patristic distances between the viral haplotypes isolated from the patients (or alternatively between consensus sequences when haplotypes are not available). The model, implemented as a reproducible pipeline was validated using data from a prospective dengue cohort from Sri Lanka (the Colombo Dengue Study - CDS). Full code and documentation for DENcode are available at [https://github.com/DENVGenomics/DENcode].

## Results

### Model validation

To assess the robustness of the model to parameter uncertainty, parameter variation (see methods) was evaluated across 100 Monte Carlo runs sampling mosquito mortality, spatial decay, and infectiousness. The distribution of $P_{ij}$ estimates for each case pair was summarised using the width of the 90% credible interval. (Table 1) Across all 1,298 links, the median interval width was $6.0 \times 10^{-8}$. Only 5% of links showed widths greater than 0.017, and the largest observed width (0.21) still represents a relatively narrow uncertainty range on a 0–1 probability scale, providing a measure of stability in the estimated linkage probabilities across parameter fluctuations.

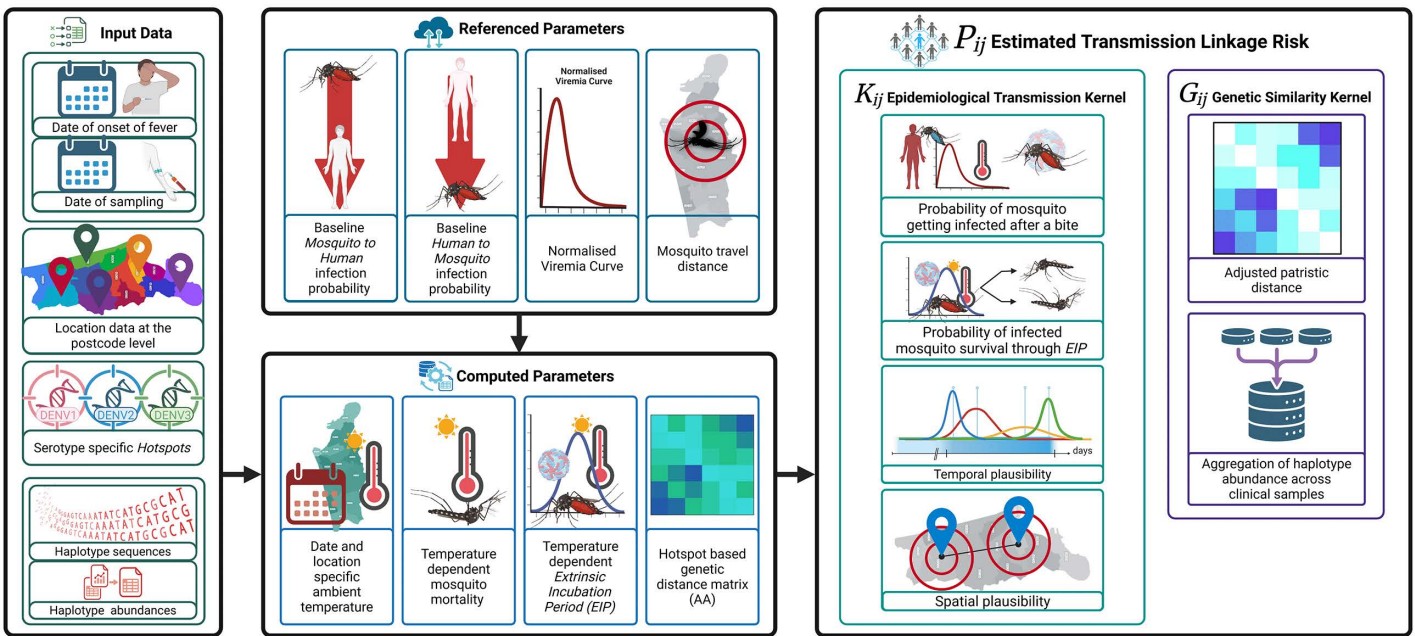

**Fig 1. DENcode is a probabilistic modelling framework that integrates haplotype level genetic data, epidemiological metadata, and vector-related variables to infer probabilistic dengue transmission networks.** Haplotype sequences are used to compute serotype-specific genetic similarity while adjusting for mutation hotspots. Epidemiological data, including onset dates, sampling dates, and postcode-level locations, inform temporal and spatial kernels. Transmission probabilities between cases are estimated by combining these components into a unified probabilistic framework, producing a network of directional links that can be interrogated with centrality measures and visualised geographically. Created in BioRender. Stone, **H.** (2026) https://BioRender.com/ff2w68r.

**Table 1. Widths of 90% credible intervals for transmission probability estimates ($P_{ij}$) across 100 Monte Carlo simulations. Interval widths are shown by probability stratum, with counts, means, medians, and maxima for each group.**

| $P_{ij}$ bin (mean $P_{ij}$) | Number of edges | Mean CI width | Median CI width | Maximum CI width |
|---|---|---|---|---|
| **Very low ($<10^{-6}$)** | 633 | $5.8 \times 10^{-8}$ | $7.2 \times 10^{-12}$ | 0.000004 |
| **Low ($10^{-6} - 10^{-3}$)** | 280 | $3.6 \times 10^{-5}$ | $1.8 \times 10^{-6}$ | 0.0012 |
| **Higher ($>10^{-3}$)** | 385 | 0.013 | 0.00092 | 0.21 |

Ablation experiments further quantified the influence of model components. Removing the genetic similarity term redistributed probability mass across links, with a mean row-wise KL divergence of 1.61, while 46% of the five most probable links per case (candidate set) were preserved. Flattening the epidemiological kernel resulted in a similar level of distributional divergence (KL 1.76) but reduced stability of the candidate set, with a top-5 Jaccard overlap of 0.35. Randomising genetic distances caused perturbation (KL 8.00) while preserving a comparable overlap of 0.45 with the baseline (Table 2).

**External validation.** Directional probability of infection for all edges of the networks (above the selected thresholds) was concordant with the temporal sequence of onset of fever (from clinical data) between each pair of cases connected. See S1 Table.

In consensus sequence networks two connected pairs for DENV2 and 19 pairs for DENV3 were also detected in the corresponding serotype-specific Bayesian phylogenetic tree clusters. However, the mean patristic distances for any of these pairs were not statistically significantly lower than the average value of the entire tree. (S2 and S3 Figs and S2 Table).

**Table 2. Effect of model component removal on transmission probability estimates. Mean row-wise Kullback–Leibler divergence and top five Jaccard overlap values are shown for models without genetics, with a flattened epidemiological kernel, and with randomised genetic distances.**

| Model modification | KL divergence | Top-5 Jaccard overlap |
|---|---|---|
| Genetics removed | +1.61 | +0.46 |
| Epidemiological kernel flattened | +1.76 | +0.35 |
| Genetic distances randomised | +8.00 | +0.45 |

**Transmission linkage risk network analysis.** After 100 Monte Carlo iterations, DENcode produced stable estimates of $P_{ij}$, summarised as means, standard deviations and centiles for each case pair (S1 Table) and serotype specific transmission networks were reconstructed retaining the edges above probability thresholds of ≥0.1 and ≥0.5 (two networks per serotype). These showed no self-loops or reciprocal links, consistent with unidirectional transmission. DENV1 had the smallest network with 7 nodes, followed by DENV3 with 33 nodes and DENV2 with 48 nodes, corresponding to the input data available for each serotype (Table 3).

Regarding network properties, both DENV1 networks had comparable node coverage, network density, median edge distances, median node time lags, and mean edge weights. The DENV2 networks showed comparable median edge distances, median time lags at both thresholds but node coverage and density were lower, and the mean edge weight was higher at the 0.5 probability threshold network as expected. DENV3 networks also had a similar pattern but notably the median node time lag was shorter than DENV1 and DENV2 networks, and more so at the 0.1 probability threshold. (Table 3).

The network centralities were calculated for each serotype at both probability thresholds. The top ranked nodes per centrality measure are summarised in Table 4.

DENV1 networks at both probability thresholds were compact with distinct functional specialisation between two dominant nodes (Fig 2). Node 800 consistently led PageRank and Eigenvector centralities. In contrast, node 601 dominated structural connectivity measures, achieving the highest total degree (5 linkages), closeness and betweenness. At the highest threshold (≥0.5) all five active nodes remained connected, but the out-degree became uniformly distributed with single linkages each, indicating that high-confidence transmission risk dispersed across multiple potential sources. DENV2 patients formed the largest network, which substantially decreased in size and connectivity as the probability threshold increased (Table 3). Node 761 exhibited multi-centrality dominance, leading simultaneously the degree-based and path-based centrality measures at both thresholds. Additionally, node 573 also led some centralities at the lower threshold but its significant dropped at the higher threshold. The network had 3-core connectivity at ≥0.1 which decreased to 1-core at ≥0.5 probability threshold. DENV3 had an intermediate sized network (25 nodes, 41 probabilistic linkages at ≥0.1 versus 22 nodes, 19 linkages at ≥0.5), with two dominant nodes (640, 728) for most centralities across both probability thresholds. The network achieved 3-core connectivity at ≥0.1 which decreased to 1-core at ≥0.5 probability threshold.

**Table 3. Summary statistics of the DENV1 DENV2 and DENV3 serotype specific networks at ≥0.1 and ≥0.5 thresholds.**

| Serotype | Threshold | Total Edges | Active Nodes | Total Nodes | Node Coverage | Network Density | Mean Edge weight | SD of edge weight | Median Edge Distance (Km) | Median Node Time Lag (Days) |
|---|---|---|---|---|---|---|---|---|---|---|
| DENV1 | ≥0.1 | 6 | 6 | 7 | 85.7% | 0.20 | 0.800 | 0.315 | 107.5 | 295.16 |
| | ≥0.5 | 5 | 6 | 7 | 85.7% | 0.17 | 0.940 | 0.038 | 107.7 | 295.16 |
| DENV2 | ≥0.1 | 87 | 44 | 48 | 91.7% | 0.05 | 0.449 | 0.177 | 4.6 | 465.71 |
| | ≥0.5 | 25 | 27 | 48 | 56.3% | 0.04 | 0.638 | 0.184 | 5.3 | 437.05 |
| DENV3 | ≥0.1 | 41 | 25 | 33 | 75.8% | 0.07 | 0.504 | 0.362 | 5.4 | 43.29 |
| | ≥0.5 | 19 | 22 | 33 | 66.7% | 0.04 | 0.864 | 0.199 | 4.3 | 99.88 |

**Table 4. Top-ranked nodes by centrality measure across DENV serotypes and thresholds.**

| Centrality Measure | DENV1 (≥0.1) | DENV1 (≥0.5) | DENV2 (≥0.1) | DENV2 (≥0.5) | DENV3 (≥0.1) | DENV3 (≥0.5) |
|---|---|---|---|---|---|---|
| **Network Size\*** | 6 nodes, 6 edges | 6 nodes, 5 edges | 44 nodes, 87 edges | 27 nodes, 25 edges | 25 nodes, 41 edges | 22 nodes, 19 edges |
| **In-Degree\*** | 601 (4) | 601 (4) | 761 (35) | 761 (17) | 640 (9) | 640, 728 (6) |
| **Out-Degree\*** | 419 (2) | 124, 419, 557, 601, 665 (1) | 573 (4) | 139, 144, 159, 172, 179+ (1) | 504, 532, 793 (4) | 452, 453, 505, 513, 540+ (1) |
| **Total Degree** | 601 (5) | 601 (5) | 761 (35) | 761 (17) | 640 (10) | 640 (7) |
| **PageRank** | 800 (0.364) | 800 (0.361) | 761 (0.273) | 761 (0.315) | 728 (0.253) | 728 (0.246) |
| **Betweenness** | 601 (0.150) | 601 (0.200) | 573 (0.060) | 498 (0.002) | 640 (0.005) | 640 (0.014) |
| **Closeness** | 601 (0.800) | 601 (0.800) | 761 (0.804) | 761 (0.654) | 728 (0.426) | 728 (0.381) |
| **Eigenvector** | 800 (1.000) | 800 (1.000) | 761 (1.000) | Not available† | 728 (1.000) | 728 (1.000) |
| **Maximum Coreness\*\*** | 419, 601, 800 (2) | All nodes (1) | 13, 42, 148, 182, 186+ (3) | 139, 144, 159, 172, 179+ (1) | 452, 453, 504, 532, 564+ (3) | 452, 453, 505, 513, 540+ (1) |

\*Degree measures show connection counts. \*\*For maximum coreness the K-core values indicate maximum core membership. Values in parentheses indicate centrality scores or connection counts. Edges represent probabilistic transmission linkages *Pij* ≥ threshold). +denotes additional nodes achieved this level. † Eigenvector centrality did not converge for DENV2 at ≥0.5 threshold due to network structure.

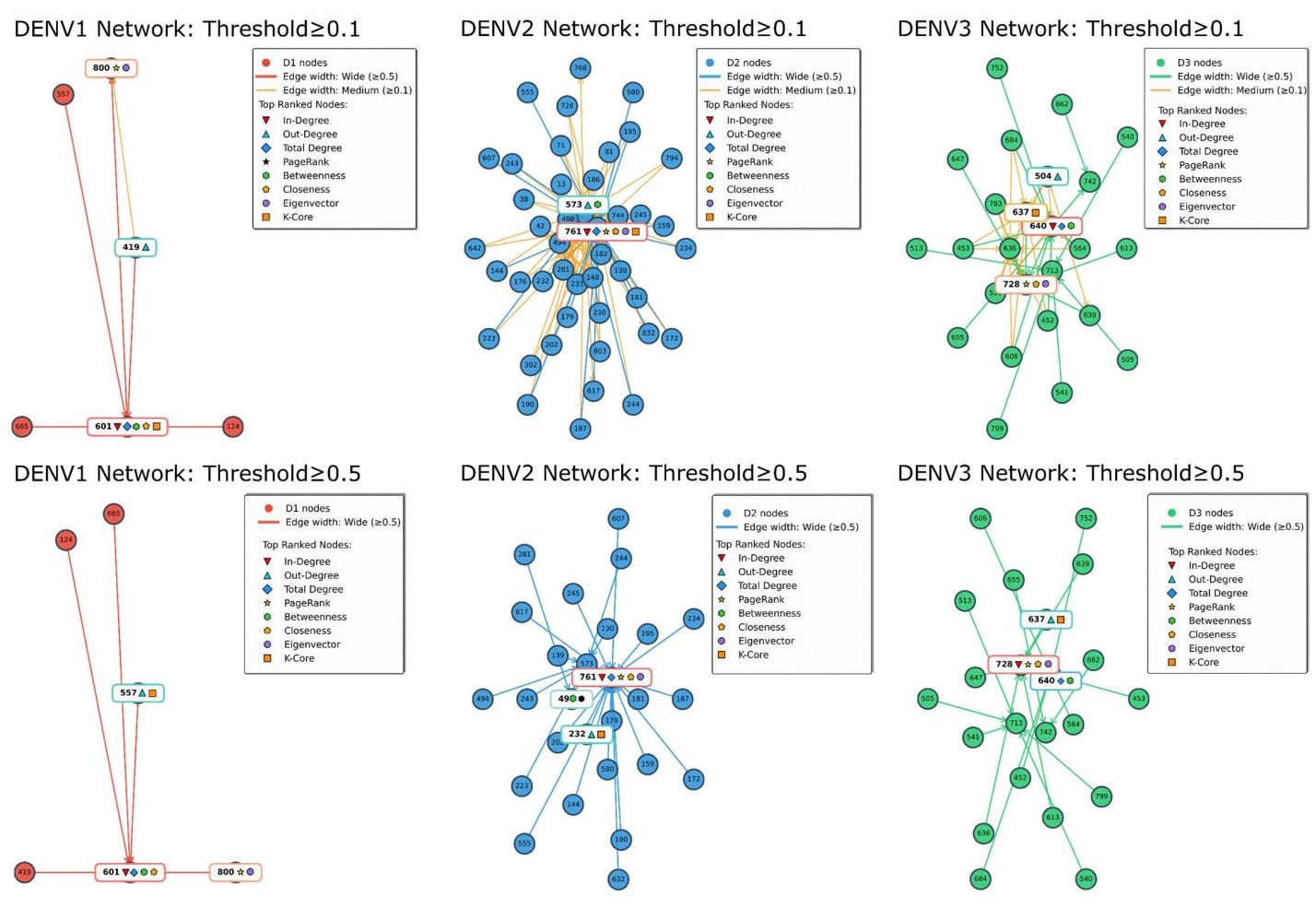

**Fig 2. Network diagrams of DENV1 DENV2 and DENV3 serotype specific networks at ≥0.1 and ≥0.5 thresholds.** Significant nodes with top centrality measures are highlighted in each network.

**Geographical mapping of nodes.** The nodes of the networks of DENV1, DENV2 and DENV3 at ≥0.1 and ≥0.5 thresholds were mapped to the centroid location of the postcode where the patient was resident at the time of infection (Fig 3). Since CDS patient recruitment was from a hospital in Colombo district in Sri Lanka most locations mapped to post codes within the same district for all 3 serotypes as expected. However, node 601 in DENV1 network which led its closeness and betweenness centralities mapped to a city (Galle district) 100km away from Colombo.

### Intersection analysis

When DENV2 (n = 44) and DENV3 (n = 37) networks were rebuilt with consensus sequences and compared to haplotype level networks at ≥0.1 and ≥0.5 thresholds (Table 5), consensus-based inference showed poor topological agreement (Jaccard <0.15) and substantial edge loss (39–100%; Fig 4A). At the ≥0.1 threshold, consensus identified only 38% of haplotype edges (49/128), which further fell to 5% at ≥0.5 threshold (2/44). For DENV2 at ≥0.5, consensus networks failed to identify any transmission links.

For common edges identified by both approaches (n = 8–12 at ≥0.1; Fig 4C), haplotype and consensus $P_{ij}$ values showed moderate-to-strong correlations (r = 0.66-0.79, p < 0.05; Fig 4D-4F) but substantial absolute differences (MAE = 0.40-0.59). Correlation scatter plots revealed that consensus networks had preserved relative ranking (diagonal trend) but systematically underestimated magnitudes (regression line below identity; Fig 4D-4F). Statistical comparison of all-edge $P_{ij}$ distributions confirmed this systematic bias (consensus mean 0.21-0.40 vs haplotype 0.60-0.98, mean difference = -0.40 to -0.59, p≤0.001; Fig 4G and Table 5)

### Discussion

We developed DENcode a theoretical probabilistic framework to infer serotype specific probabilistic dengue transmission networks and validated it using clinical, and genetic data from a hospital-based patient cohort in Colombo, Sri Lanka. Exploration of the DENcode output revealed serotype-specific differences in transmission network topology and connectivity, reflecting the epidemiological context of dengue transmission in Colombo, Sri Lanka during the data collection period (2017 – 2020), where DENV1 represented background endemic transmission, while DENV2 and DENV3 infections were part of two consecutive outbreaks in 2016–2018 (DENV2), and 2018 – 2020 (DENV3) [17].

For CDS data, DENcode revealed additional information on transmission dynamics compared to the stand-alone phylogenetic analyses published previously [17]. In Sri Lanka, circulation of all four DENV serotypes is endemic. However, any outbreaks of disproportionate magnitude that occur on top of the background circulation is typically dominated by one serotype. The period of sampling in CDS (2017 – 2020) captured the transition between two such outbreaks where an DENV2 outbreak on decline (2016–18) and a DENV3 outbreak was on the rise (2018–2019). In contrast the last known DENV1 outbreak in Sri Lanka was during 2008–10, and hence the few DENV1 cases detected in CDS probably represented background endemic transmission. Thus, it was likely that the few DENV1 infections captured in CDS had a higher transmission linkage probability than any two random DENV2 or DENV3 cases which were identified in larger numbers during the sampling period. This was reflected in the stability and consistent coverage of the smaller DENV1 network output from DENcode at both probability thresholds. On the other hand, DENV2 and DENV3 networks which were larger in size had many low probability connections resulting in loss of network coverage and active nodes when the higher probability threshold was applied. In DENV3 networks the median node time lag was shorter compared to the other serotypes probably due to the rapid increase in DENV3 cases at the time.

Interpretation of network properties was performed primarily within individual serotypes rather than through direct quantitative comparison across serotypes. In particular, the DENV1 network was derived from a substantially smaller number of cases and likely represents background endemic transmission rather than an outbreak-driven network. As a result, differences in network size, density, and centrality measures across serotypes reflect both epidemiological context and sample size effects and should not be interpreted as direct measures of relative transmission intensity between serotypes.

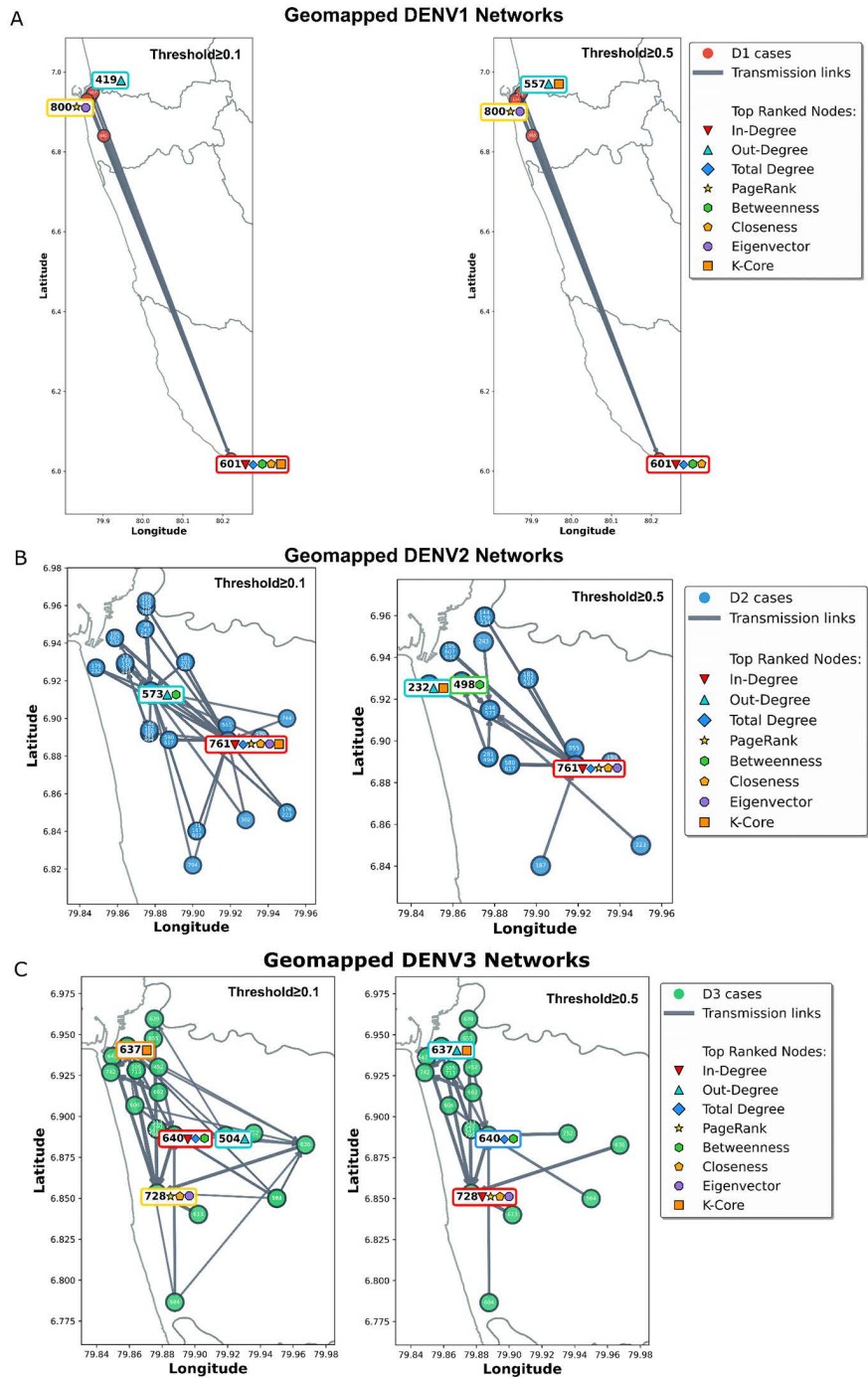

**Fig 3. Geographical network diagrams of (A) DENV1, (B) DENV2 and (C) DENV3 at ≥0.1, and ≥0.5 thresholds.** Some nodes overlap due to shared coordinates. CDS recruited patients from a single hospital in Colombo District, and this is reflected in all networks with the exception of one node in the DENV1 network which mapped outside of Colombo district with high probability. Administrative boundaries (Sri Lanka, ADM2) were obtained from geo-Boundaries [16] accessed via QuickMapTools download interface, and used under CC BY 4.0 (https://github.com/wmgeolab/geoBoundaries/raw/main/releaseData/gbOpen/LKA/ADM2/geoBoundaries-LKA-ADM2.geojson).

**Table 5. Intersection analysis of haplotype level and consensus level transmission networks at high probability thresholds.**

| Serotype | Threshold | Edges (C/H) | Common Edges (n) | Jaccard Similarity | Pearson (r) | MAE | Mean $P_{ij}$ (C/H) | Δ Mean$P_{ij}$ | P-value |
|---|---|---|---|---|---|---|---|---|---|
| DENV2 | ≥0.1 | 24/ 87 | 12 | 0.121 | 0.664** | 0.397 | 0.205/ 0.601 | -0.396 | 0.001** |
| DENV2 | ≥0.5 | 0/ 25 | 0 | 0.000 | — | — | —/ — | — | — |
| DENV3 | ≥0.1 | 25/ 41 | 8 | 0.138 | 0.787* | 0.586 | 0.397/ 0.984 | -0.586 | <0.001*** |
| DENV3 | ≥0.5 | 2/ 19 | 2 | 0.105 | — | 0.244 | 0.756/ 1.000 | -0.244 | 0.500 |

C = Consensus, H = Haplotype. Network Topology: Total edges passing threshold; **Common Edges (n)** = edges in both networks; **Jaccard** = proportion of shared edges relative to total unique edges. Edge Weight Agreement: **Pearson r** = correlation of $P_{ij}$ values for common edges; **MAE** = Mean Absolute Error for common edges. Systematic Bias: **Mean $P_{ij}$** calculated across all edges in each network; **Δ Mean $P_{ij}$** = consensus - haplotype (negative = underestimation); P-value from independent samples t-test comparing serotype- and threshold-specific mean $P_{ij}$ between consensus and haplotype networks. Significance: *** $p < 0.001$, ** $p < 0.01$, * $p < 0.05$. Decision thresholds: Jaccard >0.5 = moderate-to-good agreement; Pearson r > 0.7 = strong; MAE < 0.2 = good agreement. Pearson r undefined for n = 2; no consensus edges at DENV2 ≥ 0.5 threshold.

When the larger DENV2 and DENV3 networks were further interrogated using centrality measures, few dominant nodes emerged at the high probability threshold (e.g., node 761 ranking highest in Page rank and closeness centralities in DENV2, node 728 ranking highest in PageRank, closeness and Eigenvector centralities in DENV3). This implies that these patients were critical in linking two or more distinct infection clusters and therefore were closer transmission-wise to a larger number of patients within their respective networks. However, these assumptions can only be externally validated by additional data (e.g., person-specific movement data, or address of residence) which were unavailable for the CDS dataset. Yet, if such data were available, DENcode theoretically provides more information on current transmission dynamics to epidemiologists than traditional phylogenetics. Crude geographical mapping using the only location data available (postcode of residence at time of infection) showed that most nodes mapped within the Colombo district which is the catchment area of the hospital. Interestingly the DENV1 network had an outlier node (601) located in Galle district 100km away from Colombo which still matched to other cases of the cluster with high probability all located within Colombo. Galle is one of the few cities in Sri Lanka directly linked to Colombo by a motorway, with a travel time of one hour, resulting in substantial daily human mobility between the two cities. This illustrates the capacity of DENcode to detect probable long-distance transmission events driven by human mobility, consistent with prior observations that travel along direct routes can generate epidemiologically meaningful dispersal [17–19].

Centrality measures in DENcode-derived networks provide complementary perspectives on the potential epidemiological roles of individual cases, but should be interpreted cautiously in the context of incomplete sampling. Nodes with high out-degree or PageRank disproportionately contribute to onward transmission within the inferred network and may represent candidate transmission hubs or cases linking multiple infection chains. High betweenness centrality identifies cases that lie on many shortest paths between other nodes, suggesting a potential role as bridges between otherwise weakly connected clusters, which may reflect human mobility or spatial connectivity. In contrast, high closeness centrality indicates cases that are on average genetically and epidemiologically close to many others, placing them at central positions within the network, even if they are not highly connected by direct edges. Eigenvector-based measures further highlight nodes embedded within influential regions of the network rather than those with many direct links.

Because DENcode networks represent a probabilistic subset of true transmission events conditioned on observed cases, these metrics do not identify definitive sources or super-spreaders. Instead, they provide relative indicators of importance within the sampled transmission structure and can be used to prioritise hypotheses regarding transmission dynamics, human movement patterns, or locations where targeted surveillance or vector control may be most informative. Interpretation of network measures should therefore focus on relative patterns within a serotype-specific network rather than absolute comparisons across datasets or populations.

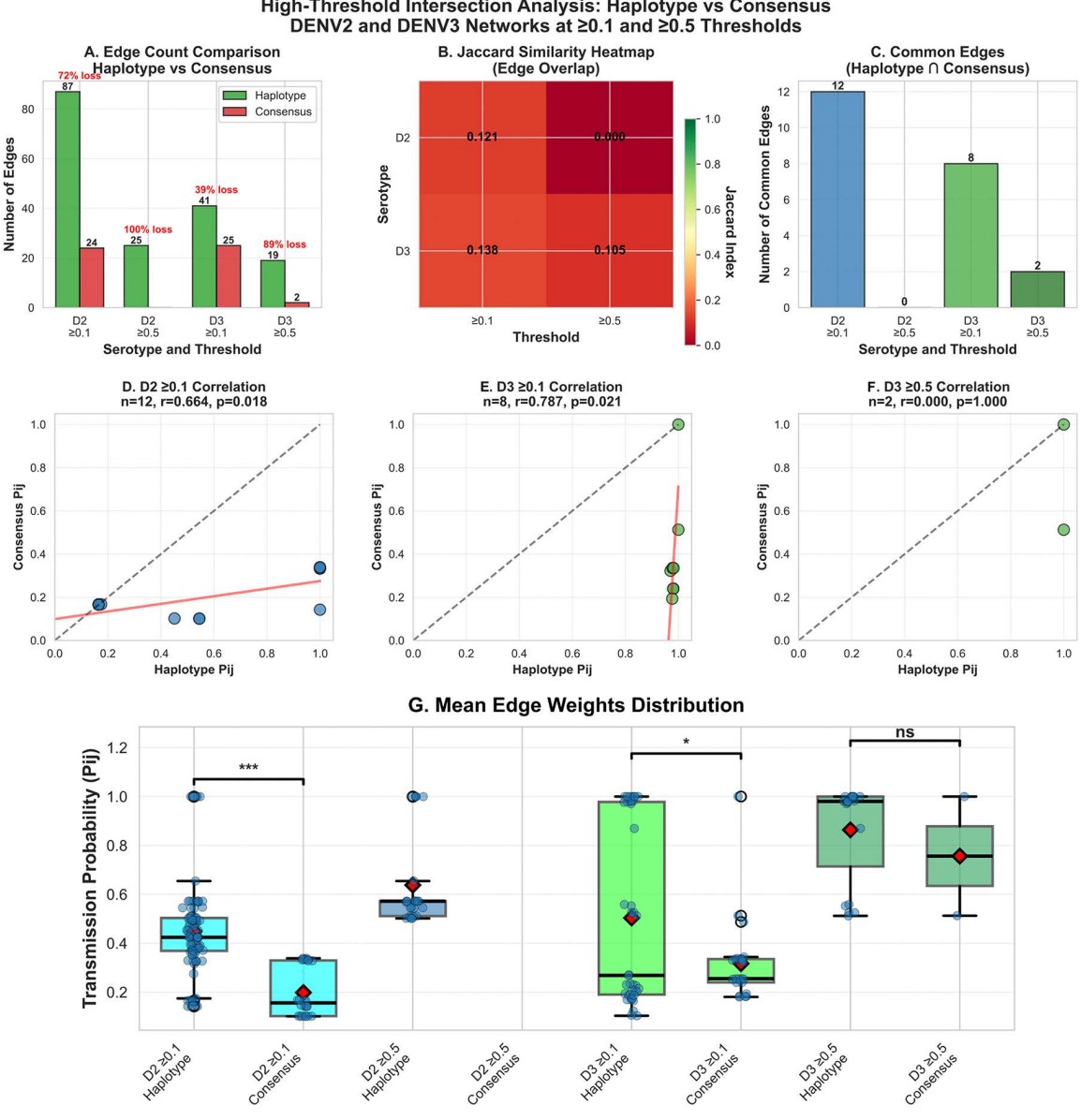

**Fig 4. Haplotype vs consensus network intersection analysis at ≥0.1 and ≥0.5 thresholds. (A)** Comparison of total edge counts showing 39-100% loss in consensus networks. **(B)** Jaccard Similarity heatmap revealing poor agreement (0.000-0.138). **(C)** Comparison of common edge counts. **(D-F)** $P_{ij}$ correlation plots for common edges of haplotype and consensus networks **(G)** Box plots comparing mean edge weights ($P_{ij}$) for haplotype and consensus networks for DENV2 and DENV3 at ≥0.1 and ≥0.5 threshold.

Model validation analyses indicated that DENcode's probabilistic estimates remained stable across 100 Monte Carlo iterations, with narrow credible intervals (median width <0.001) and consistent top-ranked transmission pairs. Sensitivity analyses through ablation experiments further showed that removing genetic or epidemiological components substantially altered the distribution of linkage probabilities (KL divergence 1.6–8.0), confirming that each term contributed meaning-fully to the inferred transmission structure. Some elements of DENcode output could also be externally validated by data

available to us. The timeline of infections as determined by the date of onset of fever was available from the clinical data linked to each case ID and this was always concordant with the higher directional probability of transmission between two cases (from an earlier infection to a latter one).

Comparison of haplotype versus consensus-based networks revealed substantial information loss when within-host diversity was reduced to a consensus. Consensus networks systematically underestimated transmission probabilities, leading to loss of edges and misidentification of network hubs, confirming prior reports that consensus sequences may obscure key transmission pathways [20]. These findings highlight the value of haplotype level data in reconstructing high-resolution transmission networks and identifying individuals of epidemiological importance. If haplotype level data is unavailable, the network connectivity at the consensus level may still improve with a larger sample size, but this aspect could not be tested with the relatively modest number of DENV2 and DENV3 sequences available to us. In addition, the degree of information loss observed with consensus-based inference is likely to depend on the underlying epidemiological context and background viral diversity, and may vary across endemic and outbreak settings.

There are several limitations of DENcode and its interpretations. First, a mathematical model cannot account for all variables influencing transmission or the outlier observations of included variables. For some variables we used what is reported in the literature but did not systematically evaluate the evidence base for such estimates. To cite an example, the flight range of the vector may vary depending on other confounding factors such as ambient temperature, altitude etc. and thus a single value reported in literature may not be representative of all environments. Also, the transmission may depend on human movement as much as it depends on vector movement, but the former could not be built into the model given the high variability of modes and purposes for human mobility. It may still be possible to observe the effect of human mobility post-hoc as unexpected occurrences within the network, such as the high probability nodes observed outside of expected range of the network. Secondly, DENcode performs better with haplotype data which is more difficult and costly to generate than consensus level data. This study used real-life data from CDS for the purpose of running and troubleshooting DENcode at haplotype level, but the networks did not fare well with the information loss when reduced to consensus sequences. Thirdly, while DENcode correctly adjusts the genetic similarity term for the effect of mutation hotspots, we have also shown previously that such hotspots may be outbreak specific. Thus, if adjusting for hotspots, they must be first identified for an outbreak using additional tools as described by us previously [17]. It is however possible to run DENcode without adjusting for the hotspot effect and specially if using consensus sequences alone, such adjustments may not be needed as only a few if any hotspots were detected at consensus level.

In conclusion, DENcode is a probabilistic framework to estimate the transmission linkage risk of a pair of dengue infections in the community taking into account pathogen genetic similarity plus vector and host dependent variables. In particular, the capacity to include haplotype level genetic information is a key aspect of its novelty. DENcode output is visualised as a network to infer directionality of transmission, spatiotemporal expansion of an outbreak, and the relative importance of each case to the transmission dynamics in the community. Thus, the additional information offered by DENcode can complement a more traditional phylogenetics-based or epidemiological exploration of a dengue outbreak. While designed specifically for dengue, the core pipeline (source code) of this model may be easily adapted for other mosquito-borne viral infections like Zika and Chikungunya with appropriate parameter modifications.

## Materials and methods

### Ethics statement

The collection of human data described in this paper is covered by human ethics committee approvals from the University of Colombo, Sri Lanka (EC-17–080) and The University of New South Wales, Australia (HC220706). Only non-identifiable human data was used. All patients provided informed written consent prior to recruitment and sample collection.

## Clinical Samples and data for validation

The clinical data for validation were from the Colombo Dengue Study (CDS) [21,22] which recruited clinically suspected dengue patients presenting within 96 hours since the onset of fever to a tertiary care teaching hospital (National Hospital, Colombo) in Colombo district, Sri Lanka. Colombo is the district with the highest population density [23] in Sri Lanka with a resident population of 2.3 million, and another floating population of about 1 million entering it daily for work and school [24]. The recruiting hospital was located in the city of Colombo within the Colombo district (both the district and the city have the same name) which is also the economic capital of Sri Lanka (Fig 5). The district of Colombo also reports the highest dengue incidence in Sri Lanka [15,24] with all four serotypes in endemic circulation.

CDS recruited 523 confirmed (by NS1 antigen test or RT-PCR) dengue patients between 2017 – 2020. Demographic and clinical data including the date of onset of fever, postcode of residence at the onset of fever, laboratory investigation results and clinical outcomes were recorded for each recruited patient (also followed up daily until discharge from hospital) and a blood sample was collected on admission for viral genome extraction. The extracted viral genomes were amplified using a whole genome amplicon assay and sequenced using Oxford Nanopore Technology to generate consensus and haplotype sequences [25]. The haplotype alignments were compared within- and between hosts to identify mutation hotspots in the virus genome [26]. The experimental and bioinformatics pipelines used for dengue whole genome amplification, sequencing, haplotype determination, and mutation hotspot identification are detailed in S1 File. The total validation dataset consisted of 90 subjects (DENV1: 8, DENV2: 51, DENV3: 31) in whom both the consensus and at least 2 within-host haplotype sequences per person had been identified previously (Fig 6) [26].

## Model formulation

Rather than reconstructing explicit transmission trees, DENcode estimates the relative likelihood of transmission linkages across all case pairs. A probabilistic pairwise framework is used because dengue transmission is only partially observed, with mosquito-mediated spread and incomplete sampling making deterministic reconstruction of transmission pairs unreliable.

For each potential source–recipient pair ($i$, $j$), DENcode estimates the relative probability that infection could have been transmitted from $i$ to $j$, conditional on observed epidemiological compatibility and genetic similarity. The resulting

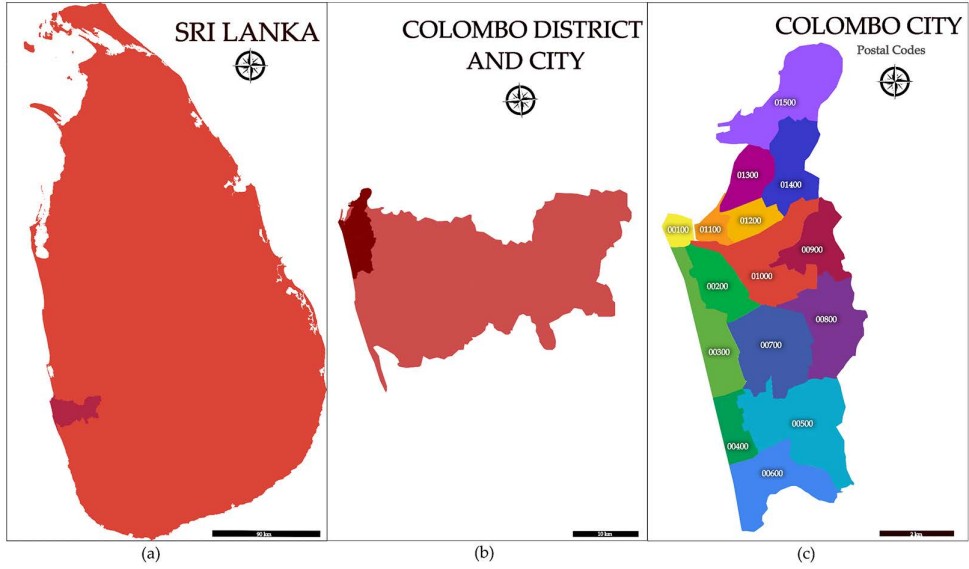

**Fig 5. Map of (a) Sri Lanka, (b) Colombo district and Colombo city and (c) the post code divisions of Colombo city.** Reproduced from [24].

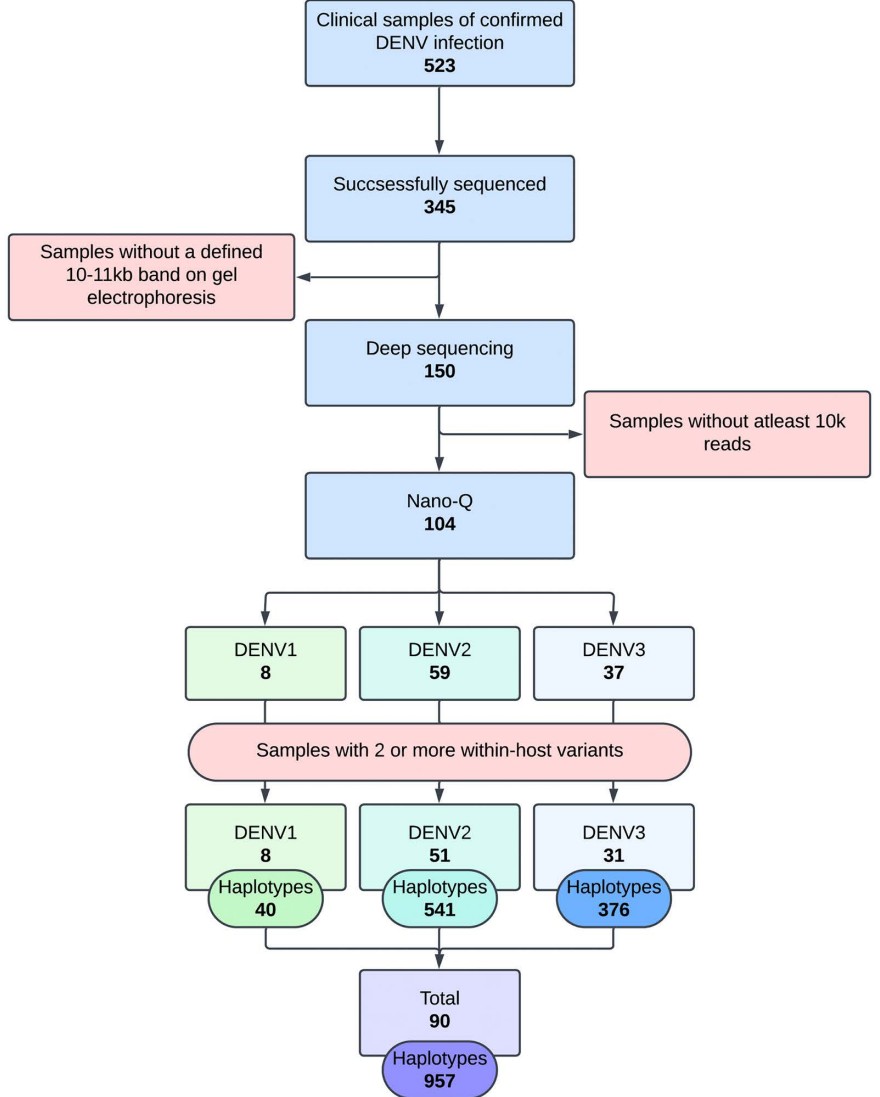

**Fig 6. Outline of sample selection for haplotype level analysis.** Adapted from Maduranga et al., *Virology*, 2026 [26].

probability is defined relative to all candidate recipients for a given source case and is therefore interpreted as a comparative measure rather than an absolute probability of direct transmission. This probability is defined as:

$$P_{ij} = \frac{K_{ij} \cdot e^{(-\gamma G_{ij})}}{\displaystyle\sum_{\substack{j' \in \mathcal{J} \\ j' \neq i}} K_{ij'} \cdot e^{(-\gamma G_{ij'})}}$$

Where $K_{ij}$ is the epidemiological kernel capturing mosquito-mediated transmission potential from case $i$ to $j$; $G_{ij}$ is the genetic divergence between viral haplotypes, measured via patristic distance. The term $e^{(-\gamma G_{ij})}$ penalises

genetic dissimilarity, with $\gamma$. Normalisation ensures that for each source case $i$, probabilities over all potential recipients $j' \neq i$ sum to 1.

The model integrates epidemiological feasibility and genetic consistency into a single transmission score for each candidate pair. The epidemiological kernel $K_{ij}$ constrains transmission to pairs that are compatible in time, space, and vector biology, while the exponential term maps genetic divergence to a continuous penalty, reducing support for links inconsistent with recent shared ancestry. These components are combined multiplicatively so that neither epidemiological plausibility nor genetic similarity alone is sufficient to produce high transmission support.

The resulting expression defines an unnormalised score for each potential transmission pair, which is then normalised across all candidate recipients $j' \neq i$ to obtain a conditional probability distribution. Consequently, $P_{ij}$ represents the relative support for transmission from $i$ to $j$ among all feasible recipients of case $i$, rather than an absolute probability of direct infection. The epidemiological and genetic components of the model are defined in detail in the following sections.

**Epidemiological Transmission Kernel.** To quantify the likelihood of transmission from case $i$ to case $j$, we define a kernel function $K_{ij}$ that models the vector-mediated transmission potential given the timing of symptoms, spatial proximity, and temperature-dependent mosquito factors such as mosquito survival and extrinsic incubation period. The kernel captures the sequential probabilities of vector-mediated transmission from an infectious human to a susceptible host, incorporating time since symptom onset, mosquito survival, and spatial proximity and can be expressed as:

$$K_{ij} = \sum_{\tau=0}^{D} I_{ij}(\tau) \cdot \lambda_i(\tau) \cdot S_{\text{mosq}}(\tau, T_i) \cdot P_{\text{mh}} \cdot f_{\text{spatial}}(d_{ij})$$

Where $\tau$ is the number of days since symptom onset in case $i$, integrated across the infectious period $[0, D]$; $t_i$ is the symptom onset date for case $i$. The indicator function $I_{ij}(\tau)$ takes the value 1 if the observed onset date for case $j$, denoted $t_j$, falls within the plausible infection window following exposure of the mosquito on day $\tau$ of case $i$'s infectious period. This window spans from $t_i + \tau + EIP(T_i)$ to $t_i + \tau + EIP(T_i) + \Delta$, where $EIP(T_i)$ is the temperature-dependent extrinsic incubation period in the mosquito and $\Delta$ is the maximum plausible intrinsic incubation period in the human host.

The term $\lambda_i(\tau) = B_h \cdot V_{(\tau)}$ models the probability that a mosquito biting case $i$ on day $\tau$ becomes infected, determined by a baseline human-to-mosquito infection probability $B_h$, and normalised viremia curve $V_{(\tau)}$, capturing the viral load kinetics, peaking shortly after symptom onset and decaying thereafter.

Following infection, the mosquito must survive long enough for the virus to complete its incubation. This survival probability is captured by:

where $\mu(T_i)$ is the mosquito's temperature-dependent daily mortality rate. Both $\mu(T_i)$ and $EIP(T_i)$ are modelled as functions of ambient temperature based on entomological studies and expressed as:

$$\mu(T_i) = 0.8692 - 0.159 T_i + 0.01116 T_i^2 - 3.408 \times 10^{-4} T_i^3 + 3.809 \times 10^{-6} T_i^4$$

$EIP(T_i) = e^{\beta_0 + \beta_T T_i}$ where $\beta_0 = 2.9$ and $\beta_T = -0.08$ [27].

$$S_{\text{mosq}}(\tau, T_i) = e^{-\mu(T_i) \cdot EIP(T_i)}$$

Where $\mu(T_i)$ is the mosquito's temperature-dependent daily mortality rate. Both $\mu(T_i)$ and $EIP(T_i)$ are modelled as functions of ambient temperature based on entomological studies and expressed as:

$$\mu(T_i) = 0.8692 - 0.159 T_i + 0.01116 T_i^2 - 3.408 \times 10^{-4} T_i^3 + 3.809 \times 10^{-6} T_i^4$$

$$EIP(T_i) = e^{\beta_0 + \beta_T T_i}$$

where $\beta_0 = 2.9$ and $\beta_T = -0.08$ [27]

The probability that a surviving infectious mosquito successfully transmits DENV to a susceptible host during a bite is denoted $P_{mh}$.

Spatial proximity between cases modulates the final transmission probability via an exponential decay function and is expressed as:

$$f_{spatial}\left(d_{ij}\right) = e^{-\delta \cdot d_{ij}}$$

Where $d_{ij}$ is the geodesic distance between residential centroid zip coordinates of cases $i$ and $j$ and delta is a spatial decay parameter reflecting the limited range of *Aedes aegypti* and the tendency for transmission to occur over short distances (Table 6).

$\beta_h$ is the baseline probability that a given mosquito becomes infected after biting an infectious human on a given day This probability is modulated by the level of viremia in the infected host, represented by $V_{(\tau)}$, which peaks at day 2 of infection and then decays. The baseline probability of a mosquito infecting a human on a given day is estimated by $P_{mh}$. The parameter δ denotes a zone of 500m around each of the case locations and that decays with the distance that is used to model the probability of cases that tend to cluster within a tight geographic area due to the limited mobility of infected mosquitoes.

**Genetic similarity kernel.** Only the coding region of viral genome was used to derive the genetic similarity kernel, as non-coding regions contribute minimally to amino acid-level divergence. To restrict candidate transmission links to genetically plausible pairs, we computed a pairwise patristic distance matrix to derive a weighted divergence score from nucleotide haplotype sequences. Maximum Likelihood phylogeny were generated using iqtree2 [33] with model selection via ModelFinder [34] GTR+F+I+R3, branch support:1000 ultrafast bootstrap [35] replicates and 1000 SH-aLRT replicates.

To account for the disproportionate influence of rapidly mutating codons, we adjusted the patristic distances for non- synonymous mutation hotspots. Translated amino acid haplotypes were aligned using MAFFT (**M**ultiple **A**lignment using **F**ast **F**ourier **T**ransform) [36], and two pairwise matrices were computed: one capturing genome-wide amino acid divergence and another restricted to known hypervariable codons. The hotspot-adjusted genetic divergence was then computed as:

$$d_{adj}^{(s)}\left(H_i, H_j\right) = d_{patristic}\left(H_i, H_j\right) \cdot \left(1 - \frac{m_{hotspot}^{(s)}}{m_{total}}\right)$$

Where, $d_{patristic}\left(H_i, H_j\right)$ is the patristic distance derived from a phylogenetic tree; $m_{total}$ is the number of amino acid mismatches between $H_i$ and $H_j$; and $m_{hotspot}^{(s)}$ is the number of these mismatches that occur at hotspot positions $S_s$.

**Table 6. Epidemiological parameters imported from published literature to DENcode and their references.**

| Parameter | Value | Reference |
|---|---|---|
| $\beta_h$ | 0.75 | [28] |
| $P_{mh}$ | 0.75 | [28] |
| δ | 0.006, 0.009 | [29,30] |
| EIP | 5–12 days | [9,27,31] |
| Viremia | Peak 0–2 days | [32] |
| Mosquito travel distance | 106m | [8] |

This hotspot adjustment was applied at the level of individual haplotype pairs. Subsequently, when aggregating across multiple haplotypes per host, distances were weighted by haplotype abundance to reflect within-host viral population structure. For cases containing multiple haplotypes, pairwise adjusted distances were aggregated across all haplotype combinations, weighting by haplotype abundance. Let $H_i$ and $H_j$ be the sets of haplotypes for cases $i$ and $j$, respectively, each with abundance $a_k$. The overall genetic similarity between cases was defined as:

$$G_{ij} = \sum_{H_k \in \mathcal{H}_i} a_k \cdot \left( \sum_{H_l \in \mathcal{H}_j} e^{d_{adj}^{(s)}(H_k, H_l)} \right)$$

## Validation

**Model validation and sensitivity analyses.** To assess the robustness of DENcode to uncertainty in biological parameters and model structure, we conducted sensitivity analyses using Monte Carlo parameter variation and targeted model ablation experiments. To assess the robustness of DENcode to uncertainty in biological parameters and model structure using two complementary approaches: parameter variation to account for biological uncertainty, and ablation experiments to evaluate the contribution of individual model components. Additionally, an external validation of findings was carried out by comparing with clinical and phylogenetic data available for the same subjects. These analyses quantify how sensitive inferred transmission probabilities are to specific modelling assumptions, rather than serving as validation against a known transmission ground truth.

For parameter variation, the model was embedded within a Monte Carlo simulation framework. Each run sampled values for mosquito mortality as a function of temperature, the spatial decay rate governing the effect of distance on transmission, and baseline human infectiousness. Parameters were drawn from distributions informed by published biological studies (Table 6). For each sampled parameter set, a complete transmission probability matrix $\boldsymbol{P}_{ij}$ was generated. The procedure was repeated 100 times, and for each potential case pair we calculated the mean, standard deviation, and 90% credible interval of the estimated probabilities.

We conducted ablation experiments to evaluate the contribution of individual model components. In the first experiment, the genetic similarity term ($G_{ij}$) was set to zero so that probabilities reflected only the epidemiological kernel ($K_{ij}$). In the second, the epidemiological kernel was replaced by a uniform structure, such that all temporally and serotype specific pairs were assigned equal weight, irrespective of distance, viremia, or mosquito survival. In the third, genetic similarity values were preserved in distribution but randomly reassigned among potential recipients for each source case, disrupting their alignment with true case pairs. Each modified model produced a probability matrix that was compared with the baseline model, defined as the complete, unmodified version of DENcode incorporating all components. Comparisons were summarised using two measures: the mean row-wise Kullback–Leibler (KL) divergence between the modified and baseline distributions, and the Jaccard overlap of the top five most likely transmission pairs per source case. KL divergence quantifies changes in the probability distribution of transmission links when a model component is removed. A higher value indicates substantial redistribution of probability mass, even if the highest-ranked candidates were preserved, capturing the changes in the edge-level probability structure. In contrast, the Top-5 Jaccard overlap measures the consistency of the most likely transmission pairs, reflecting node-level stability in the model's prioritisation of recipients. A higher overlap suggests that the model continued to select the same leading candidates despite perturbation.

Randomisation was applied to the genetic component to assess sensitivity to the alignment between sequence similarity and case pairing. In contrast, the epidemiological kernel was flattened rather than randomised, as its inputs are constrained by temporal ordering, incubation periods, and vector survival dynamics. Randomising these quantities would produce epidemiologically inconsistent transmission scenarios. Flattening the kernel removes epidemiological structure while preserving these constraints, providing a comparable assessment of its contribution to inferred transmission probabilities.

**External validation.** We next evaluated DENcode outputs using external validation approaches based on independent clinical and phylogenetic signals not directly used in model construction. To assess temporal consistency, we compared directed transmission probabilities ($i \rightarrow j$) inferred by DENcode with the corresponding clinical data on fever onset dates, verifying the modelled transmission direction agreed with the observed temporal sequence of infections.

We further validate the inferred transmission links phylogenetically by comparing them with previously published consensus-sequence Bayesian maximum clade credibility trees of CDS isolates using BEAST v1.10.4 (Bayesian Evolutionary Analysis Sampling Trees) [24]. The top 20 transmission pairs ranked by $P_{ij}$ for DENV2 and DENV3 were cross-referenced with the corresponding phylogenetic trees built for a different purpose previously. When both the source and recipient samples were present in the tree, we compared the mean patristic distances of these high-probability pairs with the overall pairwise patristic distance distribution. Significantly shorter distances among the top-ranked pairs would indicate that DENcode's inferred transmission links correspond to closely related viral lineages in phylogenetic space.

## Network analysis of transmission linkage risk

DENcode output for each pair of samples per serotype was visualised as a directed weighted network, where nodes represent clinical sample and each directed edge from $i$ to $j$ carries the model-derived probability $P_{ij}$, denoting the relative likelihood of transmission from source $i$ to recipient $j$. As $P_{ij}$ and $P_{ji}$ are not necessarily equal, edge directionality is retained throughout all analyses. Transmission networks were examined at probability thresholds of ≥0.1 and ≥0.5 to contrast moderate- and high-confidence transmission linkages. The lower threshold (≥0.1) captures a broader set of epidemiologically plausible connections and provides insight into overall network coverage and structure, whereas the higher threshold (≥0.5) restricts attention to a smaller set of highly supported links, highlighting the most confident candidate transmission events. Examining both thresholds allows assessment of network stability and core structure across different confidence levels, without reliance on a single arbitrary cutoff.

Additionally, the networks were superimposed on a geographic map with each node at the centroid of the patients' postcode at the time of infection. The mean edge distance was calculated using this mapping and similarly the mean node time lag was calculated from the node dates of infection.

General properties of each network are described using coverage and density. Specifically, network nodes with at least one connecting edge were considered as active nodes and the percentage of such nodes in a network represents "network coverage" [37]. Similarly, the Network Density, another measure of interconnectedness of the network is expressed as

$$\text{Network density} = m / (n \times (n-1))$$

where $m$ is the total number of edges and $n$ is the number of active nodes, assuming self-loops are absent [38].

The relative importance of nodes within a network were described using selected centrality measures: in-degrees, out-degrees, betweenness, closeness, eigenvector, PageRank and maximum coreness. In-degrees centrality counts the number of incoming edges of a node (higher value indicates uncertainty of the source infection) while out-degree counts the number of outgoing edges (higher value indicates transmission productivity and super-spreader events) [39]. Betweenness centrality [40] measures how often a node lies on the shortest path between other nodes and indicate critical bridges between clusters. Closeness centrality [39,41] quantifies the average distance from each node to all other nodes within the network weighted by the transmission linkage risk and is useful to identify nodes at highest risk of infection even from distant transmission networks. Eigenvector centrality [42] as implemented in this analysis determines the importance of a node by recursively weighing connections incoming and outgoing from other highly connected nodes using an undirected graph and indicate the nodes that occupy significant positions of the network structure embedded within influential pathways. PageRank centrality [43] identifies network positions that amplify onward transmission using a

directed graph by incorporating a damping factor ($\alpha = 0.85$) representing the probability of following the network structure (i.e., transmission links) versus randomly jumping to any node in the network. The algorithm normalises for the number of outgoing edges from each node, weighting the contribution of each upstream source by its own PageRank score and the number of its outgoing connections. Maximum coreness [44] identifies the largest k-core subgraph of a node where every node has at least k connections within the that subgraph and are expressed as 1-core, 2-core etc. High maximum coreness indicate membership in densely interconnected and efficient transmission cores.

### Intersection analysis

As most datasets that may use DENcode will only contain consensus sequences, we performed a sensitivity analysis by substituting haplotypes with each clinical sample's consensus genome. The model was rerun 100 times with parameter variation, generating a consensus level $P_{ij}$ matrix for each serotype. Model outputs from haplotype- and consensus level analyses were compared for identical input datasets at ≥0.1 and ≥0.5 probability thresholds for DENV2 and DENV3 (DENV1 excluded due to limited sample size).

Network topology overlap was quantified using the Jaccard similarity index, measuring the proportion of shared edges relative to the total unique edges across networks. For common edges, edge weight agreement was evaluated using Pearson correlation and mean absolute error (MAE) to assess relative ranking and absolute deviation in transmission probabilities. Pearson correlation was used to capture expected linearity between consensus- and haplotype-derived probabilities, while MAE provided a scale-independent measure of absolute difference. Differences in overall $P_{ij}$ distributions were assessed using independent samples t-tests.

## Supporting information

**S1 Fig. Process flow diagram of DENcode.** Created in BioRender. Stone, H. (2026) https://BioRender.com/urvslx0.
(DOCX)

**S2 Fig. Maximum clade credibility tree generated from cluster T.**
(DOCX)

**S3 Fig. Maximum clade credibility tree generated from cluster R.**
(DOCX)

**S1 Table. Simulated directionality with the date of onset of fever.**
(DOCX)

**S2 Table. Summary of statistical comparison of transmission pairs and non-transmission branches in the two-cluster specific MCC BEAST trees.**
(DOCX)

**S1 Text. Methods.** Reconstruction of haplotype sequences and hotspot identification.
(DOCX)

## Author contributions

**Conceptualisation:** Sachith Maduranga, Haley Stone, Chaturaka Rodrigo.

**Data curation:** Braulio Mark Valencia, Chathurani Sigera, Praveen Weeratunga, Deepika Fernando, Senaka Rajapakse.

**Formal analysis:** Sachith Maduranga, Haley Stone, Chaturaka Rodrigo.

**Funding acquisition:** Deepika Fernando, Chaturaka Rodrigo.

**Investigation:** Braulio Mark Valencia, Rowena A. Bull.

**Methodology:** Sachith Maduranga, Haley Stone, Chaturaka Rodrigo.

**Project administration:** Andrew R. Lloyd.

**Resources:** Praveen Weeratunga, Deepika Fernando, Senaka Rajapakse, Andrew R. Lloyd, Rowena A. Bull.

**Software:** Sachith Maduranga, Haley Stone.

**Supervision:** Andrew R. Lloyd, Rowena A. Bull, Chaturaka Rodrigo.

**Validation:** Haley Stone, Chaturaka Rodrigo.

**Writing – original draft:** Sachith Maduranga.

**Writing – review & editing:** Braulio Mark Valencia, Chathurani Sigera, Praveen Weeratunga, Deepika Fernando, Senaka Rajapakse, Andrew R. Lloyd, Rowena A.Bull, Haley Stone, Chaturaka Rodrigo.

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
