## [Decision Letter · Decision Letter 0]

2 Mar 2026

PCOMPBIOL-D-25-02628

DENcode: A model for haplotype-informed transmission probability of dengue virus

PLOS Computational Biology

Dear Dr. Rodrigo,

Thank you for submitting your manuscript to PLOS Computational Biology. After careful consideration, we feel that it has merit but does not fully meet PLOS Computational Biology's publication criteria as it currently stands. Therefore, we invite you to submit a revised version of the manuscript that addresses the points raised during the review process.

We look forward to receiving your revised manuscript.

Kind regards,

Xiaomin Wan

Academic Editor

PLOS Computational Biology

Benjamin Althouse

Section Editor

PLOS Computational Biology

**Additional Editor Comments:**

Please revise the manuscript point by point in accordance with the reviewers’ comments.

**Journal Requirements:**

1) We do not publish any copyright or trademark symbols that usually accompany proprietary names, eg ©,  ®, or TM  (e.g. next to drug or reagent names). Therefore please remove all instances of trademark/copyright symbols throughout the text, including:

- ® on page: 38.

2) We notice that your supplementary Figures, and Tables are included in the manuscript file. Please remove them and upload them with the file type 'Supporting Information'. Please ensure that each Supporting Information file has a legend listed in the manuscript after the references list.

3) Some material included in your submission may be copyrighted. According to PLOSu2019s copyright policy, authors who use figures or other material (e.g., graphics, clipart, maps) from another author or copyright holder must demonstrate or obtain permission to publish this material under the Creative Commons Attribution 4.0 International (CC BY 4.0) License used by PLOS journals. Please closely review the details of PLOSu2019s copyright requirements here: PLOS Licenses and Copyright. If you need to request permissions from a copyright holder, you may use PLOS's Copyright Content Permission form.

Potential Copyright Issues:

- Figures 1 and S1:  Please confirm whether you drew the images / clip-art within the figure panels by hand. If you did not draw the images, please provide (a) a link to the source of the images or icons and their license / terms of use; or (b) written permission from the copyright holder to publish the images or icons under our CC BY 4.0 license. Alternatively, you may replace the images with open source alternatives. See these open source resources you may use to replace images / clip-art:

i- Figures 3 and 5:  Please (a) provide a direct link to the base layer of the map (i.e., the country or region border shape) and ensure this is also included in the figure legend; and (b) provide a link to the terms of use / license information for the base layer image or shapefile. We cannot publish proprietary or copyrighted maps (e.g. Google Maps, Mapquest) and the terms of use for your map base layer must be compatible with our CC BY 4.0 license.

4) Please amend your detailed Financial Disclosure statement. This is published with the article. It must therefore be completed in full sentences and contain the exact wording you wish to be published.

**Reviewers' comments:**

Reviewer's Responses to Questions

**Comments to the Authors:**

Reviewer #1: This manuscript presents DENcode, a probabilistic framework that integrates epidemiological, vector-related, and viral genetic data at the haplotype level to infer dengue transmission linkage probabilities. The approach is methodologically innovative, particularly in its use of within-host viral diversity, and addresses an important limitation of consensus-sequence–based inference for acute infections. The use of real-world cohort data and comprehensive validation analyses further strengthens the study. Overall, this is a well-executed and valuable contribution. I have a few suggestions to further improve clarity and interpretation:

1. Lines 381–418: The mathematical expressions are clearly defined, but the accompanying text would benefit from further elaboration. In particular, the background motivation for each formula and the logical relationships between successive equations are not always sufficiently explicit.

2. Tables 3 and 4: The sample size for DENV1 is substantially smaller than for the other serotypes, and comparisons of network properties and centrality measures are therefore likely to be influenced by sample size effects. In addition, the rationale for selecting probability thresholds of 0.1 and 0.5 is not explicitly justified.

3. Table 5 and Figure 4: The intersection analysis clearly demonstrates information loss when using consensus sequences; however, the extent to which this conclusion generalises to larger sample sizes or different epidemiological settings is not fully discussed. Additional clarification of the scope and limitations of this finding would help avoid overgeneralisation of the results.

Reviewer #2: The authors develop a method, coined DENCode, which combines epidemiological information of Dengue cases with genetic information of the viral samples taken from these cases. The method constructs a transmission network based on a set threshold in the relative likelihood of transmission from on case to the other.

It's well-written and an interesting paper to read. In my opinion, there are a couple of points that could be addressed to improve the clarity of the paper, particularly concerning the interpretation of the results.

-Validation vs Sensitivity analysis

The authors provide a comprehensive validation of the model, both as internal and external validation. The internal validation sounds to me, however, more as a sensitivity analysis (as it is presented in the ,methods section), testing the influence of the different model and parameter choices. This sensitivity analysis is important, as the model is needs relatively many parameters to calculate a relative likelihood. However, for internal validation to what extend the results rely on the exact signals in the input data. This could for instance be done by altering or omitting parts of the data.

As part of the sensitivity analysis, the authors randomise the genetic distance information, revealing a strong influence of the genetic information on the resulting network (given the KL divergence of +8.00). However, when considering the epidemiological information, they only tested flattening the epidemiological kernel. Why not randomise the epidemiological information as well?

Network analysis

The author present a large array of network statistics about the resulting transmission networks, including metrics like PageRank and Betweenness. Although the authors explain what these metrics measure, they do not further discuss the implications for the transmission network. Why is e.g. betweenness important to measure, and what does it tell the reader about the cases with high/low betweenness? Does this inform us about potential control strategies, likely important modes/places of transmission, or something else?

Given that the network is based on a very small fraction of the true complete transmission chain (given the selected cases in relation to the total number of cases), the interpretation of the network analysis is not straight forward. However, it forms an important part of the paper. I would urge the authors to thoroughly consider the implications of the various measured metrics and add this to the discussion.

Minor comments:

Line 92: "the virus does not ... within the human host", Although it's good to signal the difference with HIC anc HCV, and within-host evolution is not a major part of dengue evolution, some mutations do arise during the infection cycle. I would suggest rewording this sentence less strongly.

Line 398: Is the equation for the timespan correct? In the pdf there seems to be only one time defined, with EIP mentioned twice (once with (T_i), once without. Tau is also mentioned twice. Seems like some small mistake was made.

**Have the authors made all data and (if applicable) computational code underlying the findings in their manuscript fully available?**

Reviewer #1: None

Reviewer #2: Yes

PLOS authors have the option to publish the peer review history of their article (what does this mean?). If published, this will include your full peer review and any attached files.

Reviewer #1: No

Reviewer #2: No

**Figure resubmission:**
---

## [Editor Report · Decision Letter 1]

11 May 2026

Dear Dr Rodrigo,

We are pleased to inform you that your manuscript 'DENcode: A model for haplotype-informed transmission probability of dengue virus' has been provisionally accepted for publication in PLOS Computational Biology.

Best regards,

Xiaomin Wan

Academic Editor

PLOS Computational Biology

Benjamin Althouse

Section Editor

PLOS Computational Biology

Thank you for addressing the reviewers’ comments satisfactorily.

---

## [Editor Report · Acceptance letter]

PCOMPBIOL-D-25-02628R1

DENcode: A model for haplotype-informed transmission probability of dengue virus

Dear Dr Rodrigo,

I am pleased to inform you that your manuscript has been formally accepted for publication in PLOS Computational Biology. Your manuscript is now with our production department and you will be notified of the publication date in due course.

With kind regards,

Anita Estes
